# Māori and Pacific People's perspectives on Group A *Streptococcus* vaccine development and delivery in Aotearoa, New Zealand

Anneka Anderson[1], Shivani Fox-Lewis[2], Cresta-Jane Afoa-Stone[1], Monleigh Muliaumasealii[3], Tanya Heremaia[1], Annie Borland[1], Stacy-Ria Te Kurapa King[1], Rachel Webb[4], Nicole J. Moreland[5], Julie Bennett[5,6]*

1 Te Kupenga Hauora Māori, The University of Auckland, Auckland, New Zealand, 2 Department of Microbiology, LabPLUS Auckland City Hospital, Auckland, New Zealand, 3 National Hauora Coalition, Auckland, New Zealand, 4 Department of Paediatrics, Child and Youth Health, The University of Auckland, Auckland, New Zealand, 5 Department of Molecular Medicine, The University of Auckland, Auckland, New Zealand, 6 Department of Public Health, University of Otago, Wellington, New Zealand

* julie.bennett@otago.ac.nz

## Abstract

Aotearoa New Zealand experiences a disproportionately high incidence of Group A Streptococcus (Strep A) disease, particularly acute rheumatic fever (ARF), with Pacific children 80 times and Indigenous Māori children 36 times more likely to develop ARF than children of other ethnicities. This qualitative study explored Māori and Pacific Peoples' perspectives on Strep A vaccine development. Guided by Kaupapa Māori and Pacific-centred research approaches, semi-structured interviews were conducted with 29 participants, including 20 whānau (family) members and nine healthcare stakeholders. Interviews were recorded, transcribed verbatim, and analysed using a general inductive thematic approach. Three interconnected themes emerged: Perceptions of ARF, vaccine development, and vaccine delivery. These themes were grounded in cultural values such as *hauora* (wellbeing), *whanaunga-tanga* (relationships), *kotahitanga* (collective action), and *tino rangatiratanga* (sovereignty and self-determination). Participants' perspectives were shaped by lived experiences of colonisation and the recent Covid-19 vaccine rollout. Māori and Pacific-led approaches rooted in cultural knowledge systems were consistently highlighted. Participants emphasised that the current healthcare system is not fit-for-purpose for Māori and Pacific health and wellbeing and must be realigned to better reflect varying worldviews. Vaccine delivery models must be holistic and flexible, incorporating a *whānau ora* (family-centred) approach. While recognising the importance of a Strep A vaccine, participants stressed that addressing social determinants of health, such as housing, poverty, and access to health care is essential to reducing disease burden. These findings demonstrate that culturally responsive approaches are essential for successful vaccination programmes. In Aotearoa New Zealand, this

**Data availability statement:** The data underlying this study are not publicly available due to ethical and cultural considerations. Participants did not consent to public sharing of full interview transcripts, and the data contain culturally sensitive and identifiable information from small communities. De-identified excerpts are available within the manuscript. Further access to anonymised data may be considered upon reasonable request and subject to approval from the New Zealand Health and Disability Ethics Committee and relevant Māori and Pacific Governance Groups.

**Funding:** This work was supported by funding from Manatū Hauora - Ministry of Health New Zealand (contract 370383-00) awarded to NJM, AA, and RW as part of the Rapua Group A Streptococcus vaccine initiative. The funders had no role in study design, data collection and analysis, decision to publish, or preparation of the manuscript.

**Competing interests:** The authors have declared that no competing interests exist.

requires embedding Māori and Pacific leadership, perspectives and participation from design to delivery, together with strengthened Māori and Pacific health workforces.

## Introduction

Aotearoa New Zealand experiences a disproportionately high incidence of Group A *Streptococcus* (Strep A, *Streptococcus pyogenes*) disease, with rates comparable to those seen in low and middle-income countries, despite its status as a high-income country with universal (publicly funded) healthcare [1–3]. Strep A colonises the skin and throat and can result in a variety of disease manifestations including invasive and immune-mediated diseases. Acute rheumatic fever (ARF) is an immune-mediated consequence of Strep A infection, and can lead to rheumatic heart disease (RHD), which disproportionately affects young adults in at-risk communities [4].

The highest rates of Strep A and its complications are experienced by Māori (Indigenous peoples of Aotearoa) and Pacific Peoples, (a collective term representing rich diversity of Indigenous communities from various Pacific nations living in Aotearoa New Zealand) [4–6]. These inequities are attributed to colonial systems and ideologies that have historically privileged non-Indigenous populations, resulting in structural inequities that continue to negatively impact health outcomes [7] and are particularly evident in the incidence of ARF and RHD. Between 2000 and 2018, hospital admissions for ARF in children aged 5–14 years were 79.6 per 100,000 population for Pacific children, 35.9 per 100,000 for Māori, and 1.6 per 100,000 for other ethnicities [4].

Multiple factors contribute to the risk of Strep A infection, with socioeconomic conditions, including limited access to primary healthcare services and overcrowded housing playing central roles [8,9]. Effectively addressing these risks requires a comprehensive approach that includes improving housing quality and ventilation, reducing overcrowding, fostering supportive communities, raising awareness of Strep A disease, and enhancing access to primary healthcare [10]. Within this broader strategy, developing a Strep A vaccine represents a vital advancement, offering the potential to prevent severe complications, especially in populations inequitably affected [11].

Although the final form of a Strep A vaccine remains uncertain, global efforts are advancing development with notable momentum provided by national and international consortia such as the Strep A Vaccine Consortium (SAVAC), the Australian Strep A Vaccine Initiative (ASAVI) and Aotearoa New Zealand's *Rapua te mea ngaro ka tau* ("Rapua", "seeking that which is hidden") [12]. In Aotearoa New Zealand, *Rapua* is generating essential local data to help ensure any future Strep A vaccine aligns with the unique epidemiology and health contexts of local communities.

Achieving equitable and culturally safe vaccine implementation requires an understanding of how Strep A vaccination is perceived by key stakeholders, particularly Māori and Pacific communities, healthcare providers, and policymakers. Research consistently shows that engaging these groups meaningfully from the outset

strengthens community ownership and improves the effectiveness of health interventions [13]. The Covid-19 pandemic in Aotearoa New Zealand highlighted the consequences of delayed engagement, with lower early vaccine uptake among Māori and Pacific Peoples. However, uptake among communities improved significantly when trusted health, church, and community leaders led targeted vaccination efforts, highlighting the importance of early and genuine involvement of inclusive, community-led approaches grounded in trusted relationships, cultural strength, and collective responsibility [14].

As part of the *Rapua* initiative, this qualitative study explored Māori and Pacific Peoples' perspectives of the development of a Strep A vaccine, and presents their recommendations for ensuring culturally responsive, equitable processes for vaccine development and delivery.

## Materials and methods

### Ethics statement

Ethics approval for this study was provided by the New Zealand Health and Disability Ethics Committee (2022 FULL 12929). Participant information sheets explained the study's purpose and the types of questions that would be asked. Written informed consent/assent was obtained from all participants/guardians. Māori and Pacific Peoples were involved in the design and conduct of the study and Māori and Pacific Governance Groups provided study oversight.

### Methodology

This qualitative study utilised a Kaupapa Māori and Pacific-centred research methodology. Kaupapa Māori research places Māori at the centre of the research process, normalising Māori world views, language, cultural practices and ideologies through a critical de-colonising framework [15]. This approach ensures the research is intrinsically relevant and responsive to the Indigenous population it aims to benefit. Similarly, Pacific research methodologies prioritise Pacific values, relationality and community leadership, ensuring culturally safe and respectful engagement with Pacific Peoples. The broad principles of Kaupapa Māori research have been successfully applied with Pacific-focussed research in ARF and other studies [16].

Semi-structured interviews were held with Māori and Pacific whānau (families), community members and healthcare stakeholders between March and November 2023. Whānau participants were recruited by purposive heterogeneous sampling, using posters in local spaces and networks within Auckland including churches, kura (schools), kōhanga (early childcare centres) and marae (community meeting spaces) and utilising social media platforms. This approach was selected to captive diverse ethnic (particularly for Pacific Peoples) representation and diverse vaccine perspectives. It also supported accessibility, reduced barriers to participation and enabled engagement with whānau in familiar trusted environments.

Healthcare stakeholders were recruited by purposive sampling via email to recruit participants working in healthcare policy or delivery roles specifically for Māori and Pacific groups. Purposeful sampling was used for healthcare stakeholders to ensure representation from those working directly in Māori and Pacific healthy policy and service delivery.

Whānau interviews were one to two hours long. Participants could choose who was present at the interview with them, alone or with whānau members present. Interviews were conducted at a time and venue agreed upon by participants, face-to-face in person (cafes, community venue, participant's home) or online via Zoom. In accordance with the principles of Kaupapa Māori research, interviews were facilitated by ethnically concordant researchers who held proficiency in te reo Māori, Tongan, Rarotongan, Niuean and Samoan, to enable bilingual discussion to occur at participants' comfort. Interviews followed ethnically specific tikanga (cultural protocols), for example opening and closing karakia (blessings), offering koha (acknowledgements) and kai (food) and establishing whakawhanaungatanga (relationship-building, establishing roles and context) prior to commencing the interview. The interviews began with a lay-person overview of Strep A and ARF including images to provide participants with foundational understanding and orientation to the topic. The interview questions were focussed on perceptions of vaccine development and delivery. Two pilot interviews were held to refine the interview questions.

Most healthcare stakeholder interviews involved a single participant, though some chose to be interviewed with colleagues. Participants were Māori and Pacific healthcare professionals who worked in early childhood vaccination, with one New Zealand European included due to an absence of a Māori or Pacific representative from their service. Ethnically concordant researchers conducted interviews, using Kaupapa Māori principles, at times and locations chosen by participants. Sessions were around one hour and began with PowerPoint slides summarising findings from the whānau interviews. Questions focussed on participant's perceptions of these findings, including gaps, feasibility of meeting whānau values and contexts, and vaccine delivery models.

Interviews were digitally audio-recorded and observational notes taken. Recordings were transcribed verbatim. De-identified transcripts were manually coded by the research team. A coding framework was developed through collaborative discussions among the researchers based on collectively agreed patterns identified from the key words and phrases. Whānau and healthcare stakeholder participants were coded separately to explore if there were any differences in their perceptions of a Strep A vaccine development and delivery.

A general inductive thematic analysis was used to identify themes emerging directly from the coded data [17]. This approach systematically identifies themes that derive from qualitative data to develop clear links to research questions or objectives. The analysis followed several stages. First, three researchers, two Māori and one Pacific, independently analysed data within the coding frameworks. The analysis sought to identify key patterns that emerged independently from the coded data and patterns that informed the research questions. The whānau participants' data was analysed first. The healthcare stakeholders' data was then analysed which allowed the researchers to determine any differences in perceptions between these groups. The researchers met regularly to compare interpretations and resolve any differences through discussion, ensuring shared understanding and consistency in theme development. The provisional themes were shared with the broader *Rapua* research team, Māori and Pacific Governance Groups and the Scientific Advisory Board of the *Rapua* project for feedback and cultural validation. All groups endorsed the findings; therefore, no amendments were made.

## Results

A total of 29 participants were included in this study (Table 1), comprising of 20 whānau members and nine healthcare stakeholders. Of the whānau participants, 13 identified as Māori, 13 as Pacific, and six participants identified as both Māori and Pacific. Thirteen whānau participants identified as female and seven as male. The whānau group represented a wide age range, from children to kaumātua (elders).

All of the healthcare stakeholders were adults, eight identified as female and one as male. Five identified as Māori, two as a Pacific, one as Māori and Pacific, and one as New Zealand European. Stakeholders were from a variety of professions related to early childhood healthcare that included elements of vaccination (policy design, delivery, information provision) including nurses, general practitioners, community pharmacists and primary health services.

Three key themes were identified from the qualitative analysis; perceptions of ARF, perceptions of vaccine development, and perceptions of vaccine delivery (See Fig 1). These themes were not mutually exclusive, core cultural values such as hauora (wellbeing), whanaungatanga (relationships), kotahitanga (collective action) and tino rangatiratanga (Indigenous sovereignty and self-determination) underlay most themes. Systemic experiences of colonisation and more recent experiences with Covid-19 vaccines informed much of the narratives of participants, evident in many of these themes. Across all themes, there was a strong preference for community-led approaches, grounded in cultural worldviews and values. Each theme is described below.

### Perceptions of rheumatic fever

Although the interview questions did not specifically ask about ARF or RHD, participants frequently shared experiences and reflections of these illnesses. Their perceptions were shaped by both personal or whānau experiences and information from health services. Having close whānau members who had ARF or RHD was common.

**Table 1. Participant characteristics.**

| Type of interview | Pseudonym | Ethnicity | Gender | Age* |
|---|---|---|---|---|
| Whānau | Kawakawa | Māori | Female | Adult |
| | Raupō | Māori | Female | Adolescent |
| Whānau | Pōhutukawa | Māori | Female | Adult |
| Whānau | Kānuka | Māori, Tongan | Male | Adult |
| | Mānuka | Māori, Tongan | Male | Adolescent |
| Whānau | Karaka | Māori | Female | Adult |
| Whānau | Hibiscus | Tongan, Niuean | Female | Adult |
| | Frangipani | Tongan, Niuean | Female | Adult |
| Whānau | Heilala | Tongan | Female | Adult |
| Whānau | Kōwhai | Māori | Female | Adult |
| | Rimu | Māori | Male | Adult |
| Whānau | Kaiatea | Māori, Rarotongan | Female | Adolescent |
| | Auere | Māori, Rarotongan | Male | Adolescent |
| | Kauri | Māori, Tongan | Male | Adolescent |
| | Pūriri | Māori, Tongan | Male | Child |
| Whānau | Miro | Māori | Male | Adult |
| Whānau | Pua | Niuean | Female | Adult |
| | Huni | Niuean | Female | Adult |
| Whānau | Milo | Samoan | Female | Kaumātua |
| | Ifi | Samoan | Female | Adult |
| Healthcare stakeholder | Kōkako | Māori | Female | Adult |
| Healthcare stakeholder | Kererū | Māori | Female | Adult |
| Healthcare stakeholder | Tui | Māori | Female | Adult |
| Healthcare stakeholder | Korimako | Tongan | Female | Adult |
| Healthcare stakeholder | Riroriro | Māori | Male | Adult |
| Healthcare stakeholder | Kea | Māori | Female | Adult |
| Healthcare stakeholder | Kōtuku | NZ European | Female | Adult |
| Healthcare stakeholder | Huia | Māori, Rarotongan | Female | Adult |
| Healthcare stakeholder | Kākā | Samoan | Female | Adult |

*Child <13 years, Adolescent 13–20 years, Adult 21–60 years, Kaumātua >60 years

*"I personally know a young man who had Strep throat and rheumatic fever and he's 19 now and he's due for heart surgery when he's 21."* – Pōhutukawa, whānau, Māori, female, adult.

Whānau questioned why ARF and RHD are diseases of Māori and Pacific Peoples. Some attributed it to colonisation, systemic racism and resulting poverty and poor housing conditions. Others questioned if there were underlying genetic explanations.

*"I wouldn't want anyone with it. I don't know if you're talking with the lower economic [status], and that's why? Poor housing, the damper houses that are spreading these viruses and colds through their houses. They shouldn't be living in these cold damp houses, you know?... not in today's world."* – Kānuka, whānau, Māori and Tongan, male, adult.

Although most whānau participants were aware that it was important to treat sore throats as they could cause rheumatic fever, not many knew how this happened or that ARF could lead to RHD.

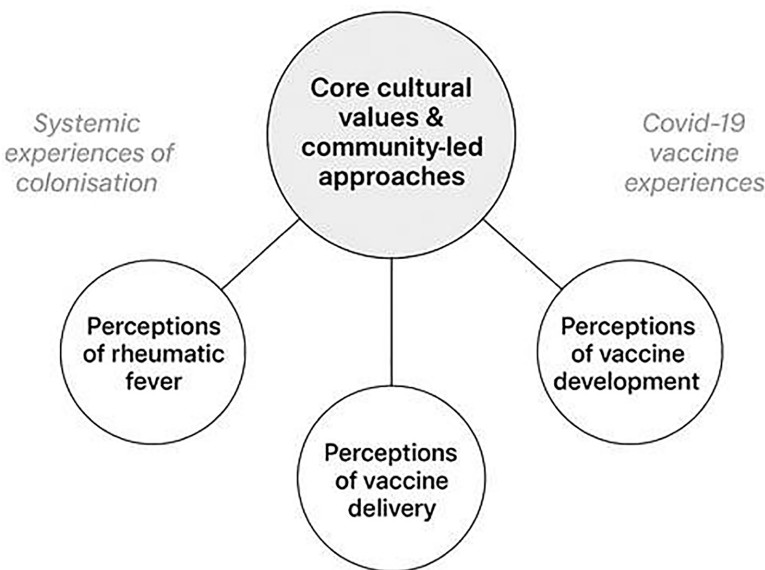

**Fig 1. Key themes of Group A Streptococcal vaccine development.**

*"But, yeah that's pretty scary when I look at it. I didn't actually realise, like I heard of rheumatic fever, but then I never really realised how serious the heart part of it was."* – Karaka, whānau, Māori, female, adult.

One participant learned about ARF through whānau experience and information shared by a trusted doctor. Across interviews, whānau highlighted the need for greater awareness, advocacy, and responsive care for sore throats, ARF, and RHD to support their hauora (wellbeing) and that of their communities.

*"... if I hadn't had this information I probably wouldn't [seek health care for children's sore throats]. I'd just go 'oh nah' until something really serious had happened."* – Pohutukawa, whānau, Māori, female, adult.

Healthcare stakeholders' perceptions of ARF aligned with those of whānau, shaped by clinical, community, and personal experiences. As with whānau, they saw colonisation and social determinants as key causal mechanisms of ARF and RHD across generations.

*"…then you've got the children coming through. So, generations of this disease... getting that family history and to demonstrate the whakapapa [genealogy] of rheumatic fever through the generation would be wonderful. So that our mokopuna [grandchildren], like 17 years from now, are not seeing the same thing happening again."* – Tui, healthcare stakeholder, Māori, female, adult.

Whānau and healthcare stakeholders identified culturally misaligned health services as significant barriers to preventing Strep A diseases. Under-resourcing in primary care and sore throat management services was highlighted as a particular concern, as was the lack of Māori and Pacific health professionals.

*"The right workforce... will either make or break it. If they're [health professionals]….not open or they're not aware of what the cultural need is…. It's like whānau are just going to be like, absolutely not going back there."* – Huia, healthcare stakeholder, Māori and Rarotongan, female, adult.

## Perceptions of vaccine development

Māori and Pacific Peoples' cultural values particularly hauora (wellbeing) and kotahitanga (togetherness) shaped participants' views on vaccine development. Experiences with Covid-19 vaccines also influenced their attitudes and understandings. The three sub-themes emerged: Benefits, concerns, and development processes.

**Perceived value of a Strep A vaccine.**  Participants widely viewed vaccines as valuable for protecting tamariki (children), mokopuna (grandchildren) and kaumātua (older adults). Even vaccine-hesitant participants recognised vaccines as a valuable addition to the *hauora kete* (array of health resources). While they might not choose vaccination for themselves or their whānau, they supported it being available as an option for ARF prevention.

*"I'm coming from the space that they [whānau], if they have choices, they make the right choices and the choices being presented to them are the right ones."* -Kawakawa, whānau Māori, female, adult.

Participants commonly said, "*prevention is better than cure*", citing historical vaccine successes like polio and smallpox. Some healthcare stakeholders highlighted the 2019 measles outbreak in Samoa, as a reminder of the serious consequences of low vaccination rates. Both groups stressed the importance of providing early, transparent information to support authentic informed consent.

*"You know, you have to really be able to have those conversations and informed consent and informed choice with those parents... talking about, 'okay, we have this, you can have this, however it doesn't mean that you're not going to get Strep A. But, you know, we can always treat it if it does arise' and all that kind of stuff."* – Kōkako, healthcare stakeholder, Māori, female, adult.

Some participants also described vaccines as a way to "*feel safe*" against getting sick and dying from infectious diseases or associated sequalae such as RHD. Young people in particular drew on their experiences of Covid-19 and other vaccines.

*"It's [developing a Strep A vaccine] a very smart idea…it makes you feel more safe and secure from the bacteria."* -Mānuka, whānau Māori and Tongan, male, adolescent.

Safety for whānau and communities was valued above individual safety for participants – particularly for protecting tamariki, mokopuna and kaumātua.

*"Yeah, anything for prevention is better than curing it so if it would prevent the virus [sic] in the first place why wouldn't you? And even if you can only save half those kid's lives, it's better than any kid, it shouldn't be any kid……And they are our most vulnerable and I wouldn't want anyone with it."* – Kānuka, whānau, Māori and Tongan, male, adult.

**Perceived concerns of a Strep A vaccine.**  A common concern for many participants was that the development of a Strep A vaccine would reallocate resources or "*mask*" primordial causes of ARF such as housing, poverty and systemic racism and that failing to address these determinants of health would maintain on-going inequities for Māori and Pacific Peoples.

*"Creating a vaccine would just mask and hide the issues that cause it……. we will still have issues with overcrowding and other illnesses; respiratory conditions and it might reduce some hospital admissions…… So, it's almost like a way of hiding the fact that we have got these really fixable issues, and they are masked."* – Ifi, whānau, Samoan, female, adult.

Another vaccine concern that emerged from whānau was vaccine side effects and not knowing what was in vaccines. For both whānau and healthcare stakeholders, trust in who and how vaccines were developed as well as information provided about vaccines was critical to these discussions.

*"I had a nephew that, he was born okay, and I think when went in for [an] injection, maybe he was five, something happened to him and he was disabled...."* – Pua, whānau, Niuean, female, adult.

Some participants viewed the health system as a colonial structure that continues to disadvantage Māori and Pacific Peoples. This mistrust, rooted in systemic racism and historical injustice, led some whānau to see vaccines as tools of colonisation. This perception could impact future engagement with healthcare, including with a Strep A vaccine.

*It's this colonisation in a different form……… our people continue to suffer……I use rongoā* [traditional Māori medicine] *as my first approach."* – Kawakawa, whānau, Māori, female, adult.

Some health stakeholders saw Strep A vaccines as a way to promote equity for Māori and Pacific Peoples. Not only by preventing ARF and RHD, but also by freeing up resources to address broader health and social inequities. Others described this as a chance to reframe vaccines as a decolonising tool, offering protection from diseases that have long disproportionately affected Indigenous communities.

*"Vaccines can be a method of decolonisation... we have forgotten the history of when colonisation came and we were devastated by disease, you know, smallpox, Spanish flu."* – Kēreru, healthcare stakeholder, Māori, female, adult.

Healthcare stakeholders highlighted that vaccines need to cover the full spectrum of Strep A diseases and should be delivered through trusted, community-based models. They highlighted the importance of Māori and Pacific leaders embedding Māori and Pacific knowledge and wisdom in the process.

*"It's about educating our whānau and having those robust kōreros (discussions)... especially in our kura (schools) and local communities, where services need to reach our most vulnerable."* – Kōkako, healthcare stakeholder, Māori, female, adult.

*"Work with pastors and churches—people families trust. That trust starts with discussion."* – Korimako, healthcare stakeholder, Tongan, female, adult.

## Perceptions of vaccine delivery

As with other themes, the core values of trust, whanaungatanga, tino rangatiratanga and āwhina (caring) underlay participants' narratives around the ways that vaccines should be delivered. Four key sub-themes emerged: Culturally aligned and flexible vaccine delivery; needle- or pain-free options; better resourcing for primary healthcare and clinical practice; and strengths-based information strategies.

**Culturally aligned models of vaccine delivery.** Participants felt that vaccine delivery needed to be under Māori or Pacific governance, involving trusted providers and grounded in cultural values to build trust and rapport with communities.

*"Cause through church as well, you know, we look after our members…….We have that open communication, where they ask us 'is there anything that you guys want us to do?'"* – Pua, whānau, Niuean, female, adult.

*"The most important [thing] is to keep the foundation strong and give that comfortable space that people from the Islands are more familiar with and they will come"* – Milo, whānau, Samoan, female, kaumatua.

Participants strongly supported flexible, community-based vaccine delivery models; such as home visits, mobile clinics, drive-throughs, and school, church, or cultural event-based services delivered through a healthy families (whānau ora) approach, where health and social services wrap around the whole whānau with support from trusted navigators.

*"Having pop up stalls with registered nurses.........it's having someone there to explain, and it's doing it right then and there."* - Raupō, whānau, Māori, adolescent.

*"It's so hard to even find the time in those early years [to get their children vaccinated], considering how many vaccines there are, how many Plunket check-ups, how many doctors check-ups, midwifery check-ups. So, having the [childhood vaccination] service designed to work for whānau, I think in this case has been working really well for us."* Rimu, whānau, Māori, male, adult.

One healthcare stakeholder raised concerns that ethnic concordance could cause mistrust in some Pacific communities, where health workers may be well known.

*"I know what you do outside of work so why would I trust you with this information?"* – Kākā, healthcare stakeholder, Samoan, female, adult.

**Needle- and pain-free vaccine delivery.** Fear of needles and vaccination pain was common across all ages and genders. Many participants supported needle-free vaccine options, with adults describing the distress for themselves and their children, and rangatahi (adolescents) finding needles scary.

*"Maybe find an alternative way for vaccines......Something without the needle"*. - Mānuka, whānau, Māori and Tongan, male, adolescent

Despite the cultural prevalence of tā moko (traditional Māori body art), tatau (traditional Pacific body art), many still feared vaccine needles. This sparked discussion about the difference between vaccine injections and traditional tattooing. Both whānau and health stakeholders explained that tā moko carries cultural meaning and connection, unlike vaccination.

*"I was told once, for someone that don't like needles why do you have lots of tattoos? But it's a different needle mate, like that".*- Miro, whānau, Māori, male, adult.

*"What they're getting [with tā moko] is something that they can relate to, like, in terms of their whakapapa (ancestry), that's coming into them. It's not something that it's not foreign, that they are unaware of into their wairua (soul)".* Kōkako, health stakeholder, Māori, female, adult.

**Limited resourcing for Māori and Pacific primary healthcare.** Participants expressed concerns that current funding restrictions could compromise future community vaccine delivery. Given ongoing workforce shortages, sustainable models would need adequate resourcing and may need to involve non-clinical kaimahi (workforce) and supportive non-Māori/Pacific providers despite preference for ethnically concordant models.

*"Our workforce is very small……. So, we need to work out who our allies are around giving and receiving information and who's at hand and who's willing to be along the journey I suppose."* – Kea, healthcare stakeholder, Māori, female, adult.

**Strengths-based information strategies.** Participants emphasised the need for culturally responsive vaccine information, tailored to different ages, languages, and literacy levels. They also stressed the importance of respectful, trust-based interactions with health professionals to avoid whānau feeling taken advantage of.

*"Sit and have a conversation and find out about each other, and you know, and then fill the gaps of their knowledge of what you know... how we're talking about Pacific People really wanting skilled and knowledgeable clinicians delivering their care, you know, making sure we're keeping our vaccinators highly skilled and educated"* – Kākā, healthcare stakeholder, Samoan, female, adult.

Discussions reflected an awareness of the power dynamics between whānau and healthcare providers, as well as within whānau themselves. Healthcare stakeholders also stressed the importance of using strengths-based, non-stigmatising language and imagery in vaccine communications.

*"Non-deficit framing and looking at this, yeah. Um, but, yeah, that kind of... That perception there is that stigmatism that it has come through, again, through some of the messaging, um, and education things that have been there in the past too. But, yeah, it absolutely is not just a disease of Māori and Pacific People."* – Riroriro, healthcare stakeholder, Māori, male, adult.

## Discussion

This study shares the perspectives of Māori and Pacific whānau and healthcare stakeholders on how to ensure equitable and culturally responsive development and delivery of a Strep A vaccine. Participants, drawing on experiences of systemic colonisation, and the Covid-19 vaccine rollout, described deep harm and mistrust in a healthcare system that often excludes different cultural values and worldviews. At the same time, there was a strong sense of motivation and empowerment to transform health systems to better align with cultural, social, and ideological contexts. These findings align with growing advocacy for authentic community involvement from the outset of any initiative, upholding the principle of *tino rangatiratanga* (Indigenous sovereignty) [18]. Meaningful engagement must be grounded in equity, cultural responsiveness, and an understanding of systemic discrimination, recognising the need for genuine inclusion to actively address the inequitable systems that persist [7].

Another key message was that while a Strep A vaccine is important, it must be delivered alongside broader efforts to address the social determinants of health, through flexible, whānau-centred, and community-led models. Transparent, decolonising processes and co-designed, well-resourced programmes were seen as essential not only for building trust, but for transforming vaccines from symbols of colonisation into tools for Indigenous health and sovereignty. It is important that future rheumatic fever prevention strategies include a multipronged approach where the vaccines are delivered alongside other primordial and primary interventions [8,9].

Recent vaccine rollouts have shown that without culturally responsive approaches, well-intentioned interventions can exacerbate inequities. Covid-19 vaccine delivery in Aotearoa New Zealand highlighted these shortcomings, with Māori experiencing the lowest vaccination rates [19]. Globally, Indigenous and marginalised groups faced similar issues where government strategies excluded community leadership, knowledge and cultural protocols [19–23], eroding trust in vaccination programmes [13,20,24]. These lessons are vital for Strep A vaccine development and delivery. Failing to centre Indigenous leadership and sovereignty not only risks perpetuating inequities but also contravenes human rights [25]. In Aotearoa New Zealand, Te Tiriti o Waitangi (the founding documented agreement between Māori and the Government) guarantees Māori tino rangatiratanga (sovereignty) over health, and health access [18,26]. Upholding Te Tiriti o Waitangi is essential for equitable, effective and culturally safe vaccine delivery. These community perspectives can directly inform the design and delivery of future Strep A vaccine programme and related policy frameworks.

 

Strengths of this study include its Kaupapa Māori and Pacific-led methodology and inclusion of community and healthcare stakeholders. Limitations include the single-centre design, that Pacific participants did not represent all Pacific ethnicities residing in Auckland, and the potential for bias due to purposeful sampling. Limited Pacific staff may have affected the depth of analysis of Pacific participants' data. In future, more in-depth studies with Pacific communities are recommended. As the study was conducted in an urban setting, findings may differ in other regions or rural settings. In future, more in-depth studies with Pacific communities and in diverse geographical settings are recommended.

These themes reflect broader global evidence showing that the legacy of colonisation, institutional racism, and the erosion of Indigenous sovereignty continue to shape vaccine confidence and engagement among Indigenous peoples worldwide. Similar calls for self-determination, culturally led design, and community governance have been documented in Indigenous health research across Australia, Canada, and the United States [27]. Positing these findings with the international literature highlights the relevance of Aotearoa New Zealand as a case study with lessons that may strengthen vaccine equity efforts for Indigenous and marginalised populations globally.

These findings are relevant to any vaccine delivery programme relevant to Indigenous and underserved communities. Acceptable immunisation programmes for Aotearoa New Zealand must embed Māori and Pacific leadership, participation, and paradigms at all stages. There is an urgent need to strengthen the workforce to shift from deprioritised to equity-based health delivery.

## Supporting information

**S1 Checklist. Inclusivity in global research.**
(DOCX)

## Acknowledgments

The authors would like to extend sincere thanks to all participants in this study. The authors would also like to acknowledge the *Rapua* Māori and Pacific Governance Groups and Scientific Advisory Board.

## Author contributions

**Conceptualization:** Anneka Anderson.

**Data curation:** Anneka Anderson, Shivani Fox-Lewis, Cresta-Jane Afoa-Stone, Monleigh Muliaumasealii, Tanya Heremaia, Annie Borland.

**Formal analysis:** Anneka Anderson, Shivani Fox-Lewis, Cresta-Jane Afoa-Stone, Monleigh Muliaumasealii, Tanya Heremaia, Annie Borland.

**Funding acquisition:** Anneka Anderson, Rachel Webb, Nicole J Moreland.

**Investigation:** Monleigh Muliaumasealii, Annie Borland, Stacy-Ria Te Kurapa King.

**Methodology:** Anneka Anderson, Nicole J Moreland.

**Project administration:** Cresta-Jane Afoa-Stone, Monleigh Muliaumasealii, Tanya Heremaia, Annie Borland, Julie Bennett.

**Resources:** Rachel Webb, Nicole J Moreland.

**Supervision:** Anneka Anderson, Nicole J Moreland.

**Validation:** Monleigh Muliaumasealii, Tanya Heremaia, Julie Bennett.

**Visualization:** Anneka Anderson.

**Writing – original draft:** Anneka Anderson, Shivani Fox-Lewis, Julie Bennett.

**Writing – review & editing:** Anneka Anderson, Shivani Fox-Lewis, Cresta-Jane Afoa-Stone, Monleigh Muliaumasealii, Tanya Heremaia, Annie Borland, Stacy-Ria Te Kurapa King, Rachel Webb, Nicole J Moreland, Julie Bennett.

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
