## [Decision Letter · Decision Letter 0]

27 Oct 2025

PGPH-D-25-02106

Māori and Pacific People’s perspectives on Group A Streptococcus vaccine development and delivery in Aotearoa, New Zealand

Dear Dr. Bennett,

Thank you for submitting your manuscript to PLOS Global Public Health. After careful consideration, we feel that it has merit but does not fully meet PLOS Global Public Health’s publication criteria as it currently stands. Therefore, we invite you to submit a revised version of the manuscript that addresses the points raised during the review process.

The manuscript has been evaluated by two reviewers, and their comments are available below.

The reviewers have raised a number of concerns that need attention. They request additional information on methodological aspects of the study such as convenience and purposive sampling, how consensus of themes was reached, if specific software was used for coding, and if interrater reliability or another form of validation was performed. They recommended including visual representations of key themes and findings. They also provided suggestions to improve the limitations section and the implications of the study.

Could you please revise the manuscript to carefully address the concerns raised?

We look forward to receiving your revised manuscript.

Kind regards,

Katherine Demi Kokkinias, Ph.D.

Staff Editor

Journal Requirements:

Additional Editor Comments (if provided):

Reviewers' comments:

Reviewer's Responses to Questions

**Comments to the Author**

1. Does this manuscript meet PLOS Global Public Health’s publication criteria?

Reviewer #1: Yes

Reviewer #2: Yes

2. Has the statistical analysis been performed appropriately and rigorously?

Reviewer #1: Yes

Reviewer #2: No

3. Have the authors made all data underlying the findings in their manuscript fully available (please refer to the Data Availability Statement at the start of the manuscript PDF file)?

Reviewer #1: Yes

Reviewer #2: Yes

4. Is the manuscript presented in an intelligible fashion and written in standard English?

Reviewer #1: Yes

Reviewer #2: Yes

Reviewer #1: Thank you for the opportunity to review this excellent and timely manuscript. This study is a strong example of qualitative research that addresses a critical equity issue in global health. The use of Māori and Pacific-centred approaches provides cultural depth and ensures that the findings are grounded in the lived realities of those most affected by Group A Streptococcus. The analysis is rigorous, and the insights are both original and actionable for policymakers and health professionals.

Suggestions for Minor Revisions:

Methods:

1. You mention using convenience and purposive sampling. Could you add a sentence to briefly explain the rationale behind these choices? For example, explaining that purposive sampling for stakeholders was used to ensure key professional viewpoints were captured would be helpful.

2. In your description of the thematic analysis, you note that three researchers analyzed the data. It would be helpful to briefly state how you reached a final consensus on the themes, as this would underscore the collaborative and rigorous nature of your analytic approach.

Discussion:

1. Your findings have clear relevance beyond New Zealand. You might consider adding a sentence or two that explicitly connects your core themes (such as the legacy of colonization on health system trust and the call for self-determination) to the broader global literature on Indigenous health and vaccine equity. This would help frame your important work as a key case study with lessons that resonate internationally.

2. The limitations section is concise and appropriate . To further strengthen it, you could consider briefly noting that the study was conducted within the Auckland urban area, and perspectives might differ in other settings. This is not a weakness of the study, but acknowledging it demonstrates a comprehensive view of the research context.

Thank you again for producing such a high-quality and impactful study. I believe it will be an important contribution to the field and look forward to seeing it published.

Reviewer #2: Thank you for the opportunity to review the manuscript entitled “Māori and Pacific People’s perspectives on Group A Streptococcus vaccine development and delivery in Aotearoa, New Zealand.”

This is an interesting and relevant qualitative study addressing an important gap in public health, particularly regarding community understanding and engagement in Group A Streptococcus (GAS) vaccine development and delivery among Māori and Pacific peoples. The topic is timely and valuable for guiding culturally informed strategies to reduce the burden of rheumatic fever and related diseases.

Abstract

The abstract is clear and informative; however, it should be shorter and include a summary of the key findings or results to provide readers with a concise overview of what was discovered through this study.

Introduction

The introduction provides useful background information. However, when the authors discuss the incidence and prevalence of Group A Streptococcus infection and compare it with the most affected regions globally (such as sub-Saharan Africa), this statement should be supported by an appropriate reference. I suggest adding:

Manuel V., Mocumbi A.O., Zühlke L. Rheumatic Heart Disease: Global Failure in Tackling a Common Killer. CJC Open (2025). doi: 10.1016/j.cjco.2025.09.011

Materials and Methods

This section is generally well described and detailed. However, it is important to clarify how the qualitative data were analyzed. The authors should specify:

Which analytical approach was used (e.g., thematic analysis, grounded theory, framework analysis);

How the coding process was conducted and whether it was manual or software-assisted;

Whether multiple researchers were involved to ensure inter-rater reliability or validation.

Clarifying these points will improve methodological transparency and reproducibility.

Results

The results are clearly written, but they would benefit from visual representation of key themes or findings. Consider adding:

A thematic map or conceptual framework summarizing the major themes identified;

A table showing representative quotations or participant statements linked to each theme.

These additions would make the findings easier to visualize and understand.

Discussion

The discussion is well done and aligns with the study objectives. The authors successfully connect their findings to broader public health implications. However, they may consider highlighting more explicitly how these community insights can inform vaccine implementation strategies or policy design in Aotearoa and similar contexts.

General Comment

Overall, this is a well-conceived and meaningful study with valuable implications for vaccine development and community engagement among indigenous and Pacific populations. Addressing the points above — especially regarding methodological clarity and result presentation — will strengthen the manuscript and enhance its contribution to the field.

**Do you want your identity to be public for this peer review?** For information about this choice, including consent withdrawal, please see our Privacy Policy

Reviewer #1: No

Reviewer #2: **Yes:** Valdano Manuel

---

## [Decision Letter · Decision Letter 1]

19 Dec 2025

Māori and Pacific People’s perspectives on Group A Streptococcus vaccine development and delivery in Aotearoa, New Zealand

PGPH-D-25-02106R1

Dear Dr Bennett,

We are pleased to inform you that your manuscript 'Māori and Pacific People’s perspectives on Group A Streptococcus vaccine development and delivery in Aotearoa, New Zealand' has been provisionally accepted for publication in PLOS Global Public Health.

Best regards,

Julia Robinson

Executive Editor

Reviewer Comments (if any, and for reference):

Reviewer's Responses to Questions

**Comments to the Author**

Reviewer #1: All comments have been addressed

Reviewer #2: All comments have been addressed

publication criteria?

Reviewer #1: Yes

Reviewer #2: Yes

3. Has the statistical analysis been performed appropriately and rigorously?

Reviewer #1: Yes

Reviewer #2: Yes

4. Have the authors made all data underlying the findings in their manuscript fully available (please refer to the Data Availability Statement at the start of the manuscript PDF file)?

Reviewer #1: Yes

Reviewer #2: Yes

5. Is the manuscript presented in an intelligible fashion and written in standard English?

Reviewer #1: Yes

Reviewer #2: Yes

Reviewer #1: (No Response)

Reviewer #2: The author has addressed all my comments appropriately. Compliments.

**Do you want your identity to be public for this peer review?** For information about this choice, including consent withdrawal, please see our Privacy Policy

Reviewer #1: No

Reviewer #2: **Yes:** Valdano Manuel
